# Evaluation of Regional Water Environmental Carrying Capacity and Diagnosis of Obstacle Factors Based on UMT Model

**Chunling Jin, Qiaoyu Guan *, Li Gong, Yi Zhou and Zhaotai Ji**

Department of Civil Engineering, Lanzhou Jiaotong University, Lanzhou 730070, China
* Correspondence: a11210303@163.com; Tel.: +86-150-3107-9825

**Abstract:** In order to promote the sustainable development of a social economy and ecology, the social–economic–natural compound ecosystem (SENCE) conceptual framework was used to construct the water environmental carrying capacity index system. Taking the Gansu section of the Yellow River basin as an example, 18 indexes were selected from the 3 subsystems of social, economic, and natural ecology. Based on the unascertained measure theory and the obstacle factor model, the comprehensive level of water environmental carrying capacity in the Gansu section of the Yellow River basin from 2015 to 2020 was empirically evaluated, and the obstacles that hindered the water environmental carrying capacity were identified and analyzed. The results showed that the comprehensive level of water environmental carrying capacity fluctuated and increased overall, and it was grade III (critical load) from 2015 to 2017 and in 2019 and grade IV (weak load) in 2018 and 2020. Considering the ranking of the obstacle degree of each index, the obstacle factors were concentrated in the natural ecological subsystem, among which the obstacle factors were the discharge of wastewater, the population density, the urbanization rate, and the water resources development and utilization rate, which should be examined in the future. The research results and methods described in this paper could provide a theoretical reference for the evaluation of water environmental carrying capacity for other rivers and lakes.

**Keywords:** water environmental carrying capacity; SENCE conceptual framework; unascertained measure theory; obstacle degree; Yellow River basin

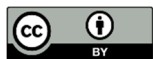

## 1. Introduction

Water environmental carrying capacity (WECC) refers to the maximum capacity of a regional water body to self-regulate and maintain sustainable social development under the requirement of specific environmental objectives [1]. With the development of a social economy, an increasing global population, and the pressure on natural ecology deepening, the conflict between a social economy and the natural ecology has been gradually intensifying, accompanied by a series of environmental problems, water pollution, soil erosion, water shortage, and frequent water logging disasters, which could cause serious harm to human society [2,3]. Reconciling the social–economic–natural ecology relationship and promoting sustainable water ecology development have been significant for river conservation and social development [4]. The Government of the People's Republic of China has elevated the ecological protection and high-quality development of the Yellow River basin to a major national strategy, advocating the protection of water security in the western part of the country and promoting ecological civilization [5]. Therefore, in order to respond to the national appeal, studying the WECC could help identify the focus points of river management, promote the improvement of the Yellow River carrying

capacity, advance ecological sustainable development [6,7], and build a beautiful Yellow River.

Currently, scholars at home and abroad have carried out different studies on the WECC. Wang [8] used the driving force-pressure-state-influence-response-management (DPSIRM) model to study the WECC of the Taihu Lake basin in terms of socio-economic impact, water resources, water quality status, and investment management. Chai [9] evaluated the WECC indicators in the Yangtze River Economic Belt (YREB) of China based on the DPSIRM. Wan [10] evaluated the WECC of the Yangtze River economic zone based on the support vector regression (SVR) model and optimized the parameters by using a cross-validation method to further improve the accuracy of the model. Song [11] calculated the WECC level of Lake Okeechobee in the United States by using a two-dimensional water environment mathematical simulation method and calculated the economic loss due to water pollution by the shadow price method. Wei [12] constructed an index system based on vigor–pressure–organization-state-resilience-management (VROSRM) and used an optimized projection tracking model to evaluate the WECC of Wuhan city. Lin [13] conducted a research analysis of water resources and WECC in the livestock industry in China based on a water footnote model. These scholars explored the WECC at different levels and revealed certain change characteristics, but there has been less research on key barrier factors, and the influence factors affecting the sustainable development of rivers should be studied.

In order to expand this field and promote the ecological sustainability of rivers, this study explored the barrier factors affecting the ecological health of rivers and provided some implications to help decision makers to manage the Yellow River. With the national strategy, the sustainable development of the Yellow River will be a long-term and continuous process. Based on sustainable development, the social–economic–natural ecological system (SENCE) conceptual framework [14,15] was proposed. This method has been used in ecological research but has not been applied to WECC; therefore, this paper adopted this method to construct the WECC index system, taking the Gansu section of the Yellow River basin as the research object, based on the improved order relation analysis-criteria importance though intercriteria correlation (G1-CRITIC) method [16,17] for subjective and objective weighting, and through the moment estimation theory [18], which reduces the interference of subjective factors, mitigates the shortcomings of the entropy method, and enhances the reliability of the results. The combination of the unconfirmed measurement and the barrier degree diagnostic models [2] for the diagnostic assessment of indicators facilitated the identification of the WECC evaluation level and major barrier factors, which provided a remedial direction and focus point for future water environment management in the Gansu section of the Yellow River basin and could have great significance for the high-quality development of the Yellow River basin.

## 2. Materials and Methods

### 2.1. Study Area

Gansu Province is located in the inland hinterland of northwest China, which is the golden route of the Silk Road in the middle and upper reaches of the Yellow River. It has a vast territory, and its population and GDP accounted for approximately 80% of the province along the Yellow River basin. It covered an area of 145,900 square kilometers, approximately 34.3% of the province's area, and the water resource content of the Gansu section of the basin reaches 17.861 billion cubic meters, accounting for 43.47% of the province's water resource content. Over the years, with population growth and economic development in Gansu, the conflict between the social economy and the natural environment has also intensified. The vast majority of the Yellow River basin in Gansu is west of the 400 mm equivalent precipitation line, the natural conditions are poor, the water function is weak, and the per capita allocation of water resources is much lower than the national average, which also resulted in a weak WECC. Its location is in western China,

information is closed, development is limited, and so economic development was relatively slow until 2021, when the gross regional product exceeded the trillion mark, reaching 1024.33 billion, accounting for about 0.9% of the country; in the country's 31 provinces and regions it ranked 27th, overall falling further backward.

In order to study the region more accurately, we selected the relevant data of the Gansu section of the Yellow River basin from 2015 to 2020 for the study, and the relevant data in the index system were selected from Gansu Provincial Development Yearbook, Water Resources Bulletin, Water Resources Yearbook, River Sediment Bulletin, Soil and Water Conservation Bulletin, Ecological and Environmental Status Bulletin, and local statistical yearbooks and water resources bulletins [2,19].

### 2.2. Constructing WECC Index System based on SENCE

### 2.2.1. SENCE Conceptual Framework

In order to develop a harmonious ecosystem and study the relationship between the social economy and natural environment, the SENCE conceptual framework involving a social–economic–natural composite ecosystem was used to construct the WECC index system, and the detailed diagram is shown in Figure 1. The social–economic–natural complex ecosystem reflects the interaction between three different subsystems: social, economic, and natural ecology; it also provides insight into whether the coexistence of human and nature are balanced [15].

The social, economic, and ecological dimensions of the system reflect the suitability of a certain area for human social activities and economic development within a certain time frame, as well as its role and impact on the evolution of movement. By coordinating its internal coupling, it supported compliance with dynamic development laws, thus promoting the sustainable development of the social–economic–natural ecological system [20,21]. It could fit the content contained in the WECC index and help to evaluate it systematically and comprehensively with objectivity, dynamism, and adjustability.

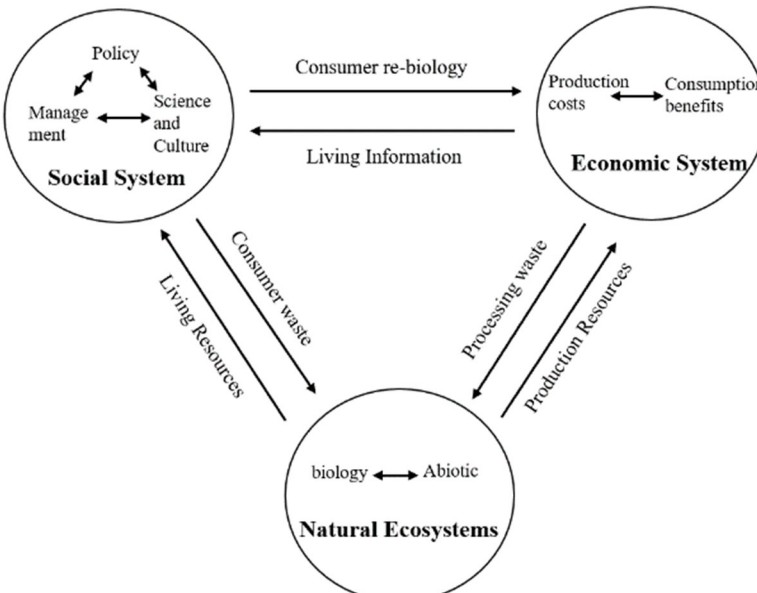

**Figure 1.** SENCE Framework.

### 2.2.2. Construction of WECC Index System

In order to study the WECC of the basin, according to the principles of science and rationality, the social–economic–natural ecological system (SENCE) conceptual framework was used to establish the index system and evaluate the data in each index. The social, economic, and natural ecological subsystems were used as the target layer of

indicators, and these three aspects were divided in detail to construct the guideline layer indicator system. Finally, 18 indicators were selected as the core of this study's indicator system, and the table of the indicator system is shown in Figure 2.

### 2.2.3. Criteria for Grading Indicators

The classification of the index level played an important role in data evaluation, and the classification criteria of the indexes in this paper were based on:

- If national or local standards for relevant indicators had been introduced, they were classified according to the national and local standards;
- If the established indicators had previous references to established standards and were more valuable for reference, the grading standards in the literature were cited;
- A statistical study area of the indicator data, according to its own data sorted by the highest or lowest value of 90% and 10%, respectively, as the standard threshold values, and then divide;
- If there was no relevant standard or reference literature, the empirical indicator grade would be used for classification. We referred to existing research results to divide the grade standard into 5 levels: heavy overload (I), overload (II), critical load (III), weakly loadable (IV), loadable (V).

The criteria for dividing the indicators in this paper are shown in Table 1. Among them, the relative values of water consumption per CYN 10,000 of industrial added value, urban sewage treatment rate, water quality compliance rate of water functional areas, water resource development and utilization rate, and soil erosion control rate were obtained according to the "Evaluation Guidelines for the Construction of Water Ecological Civilization Cities". Then population density, urbanization rate, per capita arable land area, average water consumption per mu of agricultural irrigation, investment in environmental protection as a proportion of GDP (gross domestic product), per capita water resources possession, and forest coverage rate were obtained via [6,22,23]. Finally, the grading criteria of GDP per capita, fertilizer application intensity, water consumption of CYN 10,000 GDP, energy consumption per unit of GDP, and wastewater discharge were derived from the calculation of the optimal and worst values of the data of each index in the Gansu Province of the Yellow River basin from 2015 to 2020.

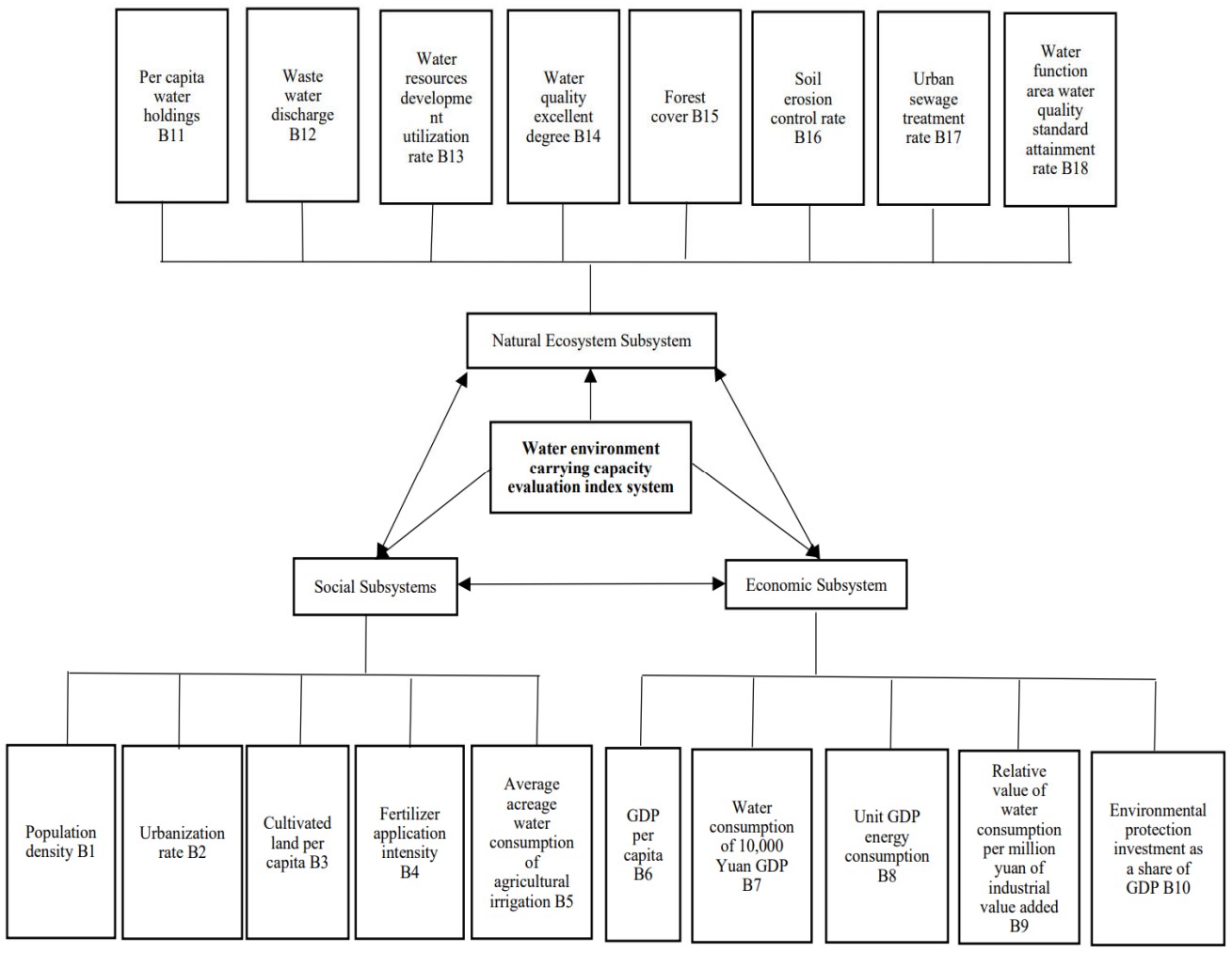

**Figure 2.** WECC index system.

**Table 1.** WECC evaluation index levels.

| Subsystems | Indicator Attribute | Indicators | Rank | | | | |
|---|---|---|---|---|---|---|---|
| | | | I | II | III | IV | V |
| Social Subsystems | $B_1-$ | Population density (persons/km²) | ≥1300 | 900~1300 | 500~900 | 100~500 | ≤100 |
| | $B_2-$ | Urbanization rate (%) | ≥75 | 60~75 | 45~60 | 30~45 | ≤30 |
| | $B_3+$ | Arable land area per capita (mu) | ≤0.8 | 0.8~1.6 | 1.6~2.4 | 2.4~3.2 | ≥3.2 |
| | $B_4-$ | Fertilizer application intensity (kg/hm²) | ≥320 | 260~320 | 200~260 | 140~200 | ≤140 |
| | $B_5-$ | Average water consumption per mu of farmland irrigation (m³/hm²) | ≥7500 | 6500~7500 | 5500~6500 | 4500~5500 | ≤4500 |
| Economic Subsystem | $B_6+$ | GDP (gross domestic product) per capita (million yuan) | ≤2 | 2~3 | 3~4 | 4~5 | ≥5 |
| | $B_7-$ | Water consumption of CYN 10,000 GDP (m³) | ≥90 | 80~90 | 60~80 | 40~60 | ≤40 |

|  | | | | | | | |
| --- | --- | --- | --- | --- | --- | --- | --- |
|  | $B_8-$ | Unit GDP energy consumption (m³) | ≥1.3 | 1.1~1.3 | 0.9~1.1 | 0.7~0.9 | ≤0.7 |
|  | $B_9-$ | Relative value of water consumption of CYN 10,000 of industrial added value (%) | ≥150 | 100~150 | 50~100 | 25~50 | ≤25 |
|  | $B_{10}+$ | Investment in environmental protection as a percentage of GDP (‰) | ≤0.5 | 0.5~1 | 1~1.5 | 1.5~2 | ≥2 |
| Natural Ecosystem Subsystem | $B_{11}+$ | Per capita water possession (m³/person) | ≤250 | 250~500 | 500~750 | 750~1000 | ≥1000 |
|  | $B_{12}-$ | Wastewater discharge (billion tons) | ≥5.5 | 5~5.5 | 4.5~5 | 4~4.5 | ≤4 |
|  | $B_{13}-$ | Water resources development utilization rate (%) | 60~100 | 50~60 | 35~50 | 20~35 | 0~20 |
|  | $B_{14}+$ | Water quality excellent degree (%) | 0~60 | 60~70 | 70~80 | 80~90 | 90~100 |
|  | $B_{15}+$ | Forest cover (%) | ≤10 | 10~20 | 20~40 | 40~50 | ≥50 |
|  | $B_{16}+$ | Soil erosion control rate (%) | 0~20 | 20~40 | 40~60 | 60~80 | 80~100 |
|  | $B_{17}+$ | Urban sewage treatment rate (%) | 0~70 | 70~85 | 85~90 | 90~95 | 95~100 |
|  | $B_{18}+$ | Water function area water quality standard attainment rate (%) | 0~40 | 40~60 | 60~75 | 75~90 | 90~100 |

*2.3. Methods*

Different weights of the indicators represented different interpretations, and they were influenced by human factors. In order to simplify the interference of subjective factors and increase the reliability of the results, this study adopted the G1-CRITIC method for subjective and objective assignments, assigned weights to their combinations by moment estimation theory, and then conducted comprehensive evaluation by unconfirmed measurement theory and introduced the obstacle diagnosis model to analyze its barriers that would have a greater impact on the evaluation objectives.

2.3.1. Indicator Weighting Methods

G1 Method

To simplify the complexity of the computational process, the assignment was based on the improved G1 method, as follows.

**Step 1:** Determine sequential relationships based on the importance of indicators

The most important index among the set of evaluation indexes $X_1, X_2, X_3...X_n$ is $X^*_1$, and the second most important index from the remaining indexes is $X^*_2$ and $X^*_1 > X^*_2 > ... > X^*_n$, in order of importance.

**Step 2:** Relative importance of adjacent indicators

A comparison based on the importance of the indicators yields:

$$\frac{w^*_{i-1}}{w^*_i} = r_i \qquad (1)$$

The table of significance quantification in Equation (1) is shown in Table 2.

**Table 2.** Importance quantification.

| Level of Importance | Extremely Important | Very Important | More Important | General Importance | Slightly More Important | Equally Important |
|---|---|---|---|---|---|---|
| ri takes the value | 2.0 | 1.8 | 1.6 | 1.4 | 1.2 | 1.0 |

**Step 3:** Calculation of indicator weighting parameters

Calculate the nth weighting parameter

$$w_n^* = [1 + \sum_{k=2}^{n}(\prod_{i=k}^{n} r_i)]^{-1} \tag{2}$$

The weight coefficients of the remaining indicators are calculated in descending order:

$$w_{i-1}^* = r_i w_i^* \ ; \ i = n, n-1, \cdots, 3, 2 \tag{3}$$

CRITIC Method

Criteria importance though intercriteria correlation (CRITIC) represents an objective weighting, which comprehensively considers the correlation between indicators and fully explores the information contained in the indicator data. It can mitigate the shortcomings in the entropy weighting method and improve the reasonableness and objectivity of the weighting [17]. Its specific steps are as follows.

**Step 1:** Dimensionless processing of indicator data

Firstly, the index data are dimensionless, and when the index is positive, it is processed using Equation (4):

$$X_{ij}' = \frac{X_j - X_j(\min)}{X_j(max) - X_j(\min)} \tag{4}$$

When the indicator is negative, it is treated using Equation (5).

$$X_{ij}' = \frac{X_j(max) - X_j}{X_j(max) - X_j(min)} \tag{5}$$

where $X_j(min)$ is the minimum value in the index $X_j$, and $X_j(min)$ is the maximum value in the index $X_j$.

**Step 2:** Calculation of index variability and correlation coefficients

$$\bar{x}_j = \frac{1}{m}\sum_{i=1}^{m} x_{ij} \tag{6}$$

$$s_j = \sqrt{\frac{1}{m}\sum_{i=1}^{m}\left(x_{ij} - \bar{x}_j\right)^2} \tag{7}$$

$$v_j = \frac{s_j}{x_j}(j = 1, 2, \cdots, n) \tag{8}$$

$$r_{ij} = \frac{\sum_{i=1}^{n}(X_i - \overline{X_i})(X_j - \overline{X_j})}{\sqrt{\sum_{i=1}^{n}(X_i - \overline{X_i})^2 \sum_{j=1}^{n}(X_j - \overline{X_j})^2}} \tag{9}$$

where $v_j$ is the coefficient of variation, and $r_{ij}$ is the correlation coefficient.

**Step 3:** Calculation of conflicting indicators

$$\eta_j = \sum_{i=1}^{n}\left(1 - r_{ij}\right)(j = 1, 2, \cdots, n) \tag{10}$$

**Step 4:** Calculate the amount of information in the index

$$C_j = v_j \sum_{i=1}^{n}\left(1 - r_{ij}\right)(j = 1, 2, \cdots, n) \tag{11}$$

**Step 5:** Calculate objective weights

$$w_j = \frac{C_j}{\sum_{j=1}^{n} C_j}(j = 1, 2, \ldots, n) \tag{12}$$

Determination of Portfolio Weights Based on Moment Estimation Theory

The subjective and objective weights were combined and integrated based on moment estimation theory to minimize the distance bias generated between them [18]. The specific process was as follows.

**Step 1:** Calculate weights corresponding to expected values

$$\begin{cases} E(w_{1j}) = \sum_{i=1}^{n} \omega_{1j} & (1 \leqslant j \leqslant m) \\ E(w_{2j}) = \sum_{i=1}^{n} \omega_{2j} & (1 \leqslant j \leqslant m) \end{cases} \tag{13}$$

where $E(w_{1j})$ denotes the subjective weight expectation, and $E(w_{2j})$ denotes the objective weight expectation.

**Step 2:** Calculate the weight importance factor

$$\begin{cases} \alpha_j = \dfrac{E(w_{1j})}{E(w_{1j}) + E(w_{2j})} \\ \beta_j = \dfrac{E(w_{2j})}{E(w_{1j}) + E(w_{2j})} \end{cases} \tag{14}$$

The final importance coefficients were as follows.

$$\begin{cases} \alpha = \dfrac{\sum_{j=1}^{m} \alpha_j}{\sum_{j=1}^{m} \alpha_j + \sum_{j=1}^{m} \beta_j} \\ \beta = \dfrac{\sum_{j=1}^{m} \beta_j}{\sum_{j=1}^{m} \alpha_j + \sum_{j=1}^{m} \beta_j} \end{cases} \tag{15}$$

**Step 3:** Calculate portfolio weights

$$W = \alpha W_1 + \beta W_2 \tag{16}$$

where $W_1$ denotes subjective weight and $W_2$ denotes objective weights.

2.3.2. Uncertainty Measure Theory Evaluation Method

Suppose the n segments of the evaluation object constitute the set S = {$S_1$, $S_2$...$S_n$}, and the m indicators of each segment constitute the space B = {$B_1$, $B_2$...$B_m$}, and the jth indicator of segment $S_i$ is represented by $a_{ij}$, and let aij have q evaluation levels, and its level is represented by the vector U = {$U_1$, $U_2$...$U_n$}. Suppose the level i + 1 is higher than the level i, $U_i + 1 > U_i$, then U = {$U_1$, $U_2$...$U_n$} would be considered as the ordered segmentation class [20].

**Step 1:** Calculate single indicator unconfirmed measures

Let $u_{ijt} = u$ ($S_{ij} \in U_t$) be the degree to which the measurement $S_{ij}$ belongs to the tth rank $U_t$, and $u$ satisfies the following conditions.

$$0 \leq u(S_{ij} \in U_t) \leq 1 \tag{17}$$

$$u(S_{ij} \in U) = 1 \tag{18}$$

$$u \big| S_{ij} \in \cup_{l=1}^{t} U_L \big| = \sum_{i=1}^{t} u(S_{ij} \in U_l) \tag{19}$$

Then, it is called an unconfirmed measure, while the single indicator measure matrix is expressed as follows:

$$(u_{ijt})_{m \times q} = \begin{bmatrix} u_{i11} & u_{i12} & \cdots & u_{i1q} \\ u_{i21} & u_{i22} & \cdots & u_{i2q} \\ \vdots & \vdots & \ddots & \vdots \\ u_{im1} & u_{im2} & \cdots & u_{imq} \end{bmatrix} \tag{20}$$

**Step 2:** Calculating multiple metric unconfirmed measures

Let $U_{in} = u$ ($S_i \in U_i$) denote the degree to which the evaluation subject segment $S_i$ belongs to performance level $U_i$, then.

$$u_{it} = \sum_{j=1}^{m} u_{ijt} \overline{W}_j \tag{21}$$

where $\bar{W}_j$ denotes the combined weight of the jth indicator, and its multi-indicator measurement matrix is:

$$(u_{it})_{n \times q} = \begin{bmatrix} u_{11} & u_{12} & \cdots & u_{1q} \\ u_{21} & u_{22} & \cdots & u_{2q} \\ \vdots & \vdots & \ddots & \vdots \\ u_{n1} & u_{n2} & \cdots & u_{nq} \end{bmatrix} \tag{22}$$

**Step 3:** Confidence identification

According to the evaluation class of the ordered segmentation class, the confidence level $\lambda$ ($0.5 \leq \lambda \leq 1$) was introduced to judge whether it conformed to the confidence criterion formula (20), so as to determine whether the evaluation object $S_i$ belongs to the $U_{tc}$ class.

$$t_0 = min\{k : \textstyle\sum_{l=1}^{k} \mu_{i1} \geqslant \lambda\}, (k = 1, 2, \cdots, q) \tag{23}$$

**Step 4:** Sequence

Sorting the ranks gives $U_1 > U_2 > ... > U_n$, and assigning values to the ranks so that $U_i = T_i$ and $T_i > T_i + 1$

$$d_{S_i} = \textstyle\sum_{t=1}^{q} T_i u_{it} \tag{24}$$

where $d_{S_i}$ is the unidentified performance index of the evaluation object $S_i$, and finally the importance ranking can be performed to discriminate its overall performance level.

**Step 5:** Performance disorder factor diagnosis

In order to study the impact of the main obstacle factors, three variables "factor deviation $I_j$", "factor contribution $F_j$", and "obstacle $B_j$" were introduced for the analysis [2]. The expression was as follows.

$$B_j = \frac{I_j \times F_j}{\sum_{j=1}^{m}(I_j \times F_j)} \times 100\% \tag{25}$$

where $F_j$ is the indicator weight value, $I_j$ is the difference between 1 and the standardized value of the indicator, and the order of influencing factors could be determined according to the ranking of $B_j$ barrier degree.

## 3. Results and Analysis

### 3.1. Results of the Weighting of Each Indicator

The values of each index were analyzed by applying the G1-CRITIC method for subjective and objective weighting, and their combinations were weighted by moment estimation theory, and their weighting results are shown in Table 3.

**Table 3.** Weighting of WECC indicators.

| Guideline Layer | Indicator Layer | G1 Method Weights | CRITIC Method Weights | Portfolio Weights |
|---|---|---|---|---|
| Social Subsystems | $B_1$ | 0.0981 | 0.1062 | 0.1024 |
| | $B_2$ | 0.0584 | 0.0990 | 0.0801 |
| | $B_3$ | 0.0298 | 0.0550 | 0.0433 |
| | $B_4$ | 0.0129 | 0.0408 | 0.0278 |
| | $B_5$ | 0.0248 | 0.0466 | 0.0365 |
| Economic Subsystem | $B_6$ | 0.0033 | 0.0399 | 0.0229 |
| | $B_7$ | 0.0019 | 0.0365 | 0.0204 |
| | $B_8$ | 0.0092 | 0.0407 | 0.0261 |
| | $B_9$ | 0.0055 | 0.0421 | 0.0251 |
| | $B_{10}$ | 0.0207 | 0.0804 | 0.0526 |
| Natural Ecosystem Subsystem | $B_{11}$ | 0.0417 | 0.0444 | 0.0431 |
| | $B_{12}$ | 0.2637 | 0.0775 | 0.1641 |
| | $B_{13}$ | 0.1569 | 0.0408 | 0.0948 |
| | $B_{14}$ | 0.1883 | 0.0370 | 0.1074 |

| | | | |
|---|---|---|---|
| $B_{15}$ | 0.0701 | 0.0885 | 0.0799 |
| $B_{16}$ | 0.0077 | 0.0475 | 0.0290 |
| $B_{17}$ | 0.0046 | 0.0407 | 0.0239 |
| $B_{18}$ | 0.0023 | 0.0375 | 0.0211 |

### 3.2. WECC Evaluation Level and Comprehensive Performance

Through the data of the 18 indicators selected in Table 1, a comprehensive analysis of its social–economic–natural ecological role combined with real-world data was performed to evaluate the WECC of the Gansu section of the Yellow River basin.

### 3.2.1. Comprehensive Performance Analysis

According to the indicator, weights and single indicator measure matrix can be derived from the multi-indicator measure value, used to show the degree of its belonging to the performance level, each indicator measure function is shown in Figure 3 below. Through the confidence guideline formula (23), the confidence level $\lambda = 0.5$ is selected and compared with the multi-indicator measurement results, so as to derive the evaluation level of the water environment carrying capacity and facilitate further analysis of the trend of the water environment carrying capacity in Gansu section of the Yellow River basin in recent years. Secondly, for a more comprehensive and reasonable evaluation of the water environment carrying capacity of the region, the performance index will be used for its comprehensive consideration, so that U1 = 1, U2 = 2, U3 = 3, U4 = 4, U5 = 5, using formula (24) for calculation, and the resulting performance index is shown in Table 4 below [24].

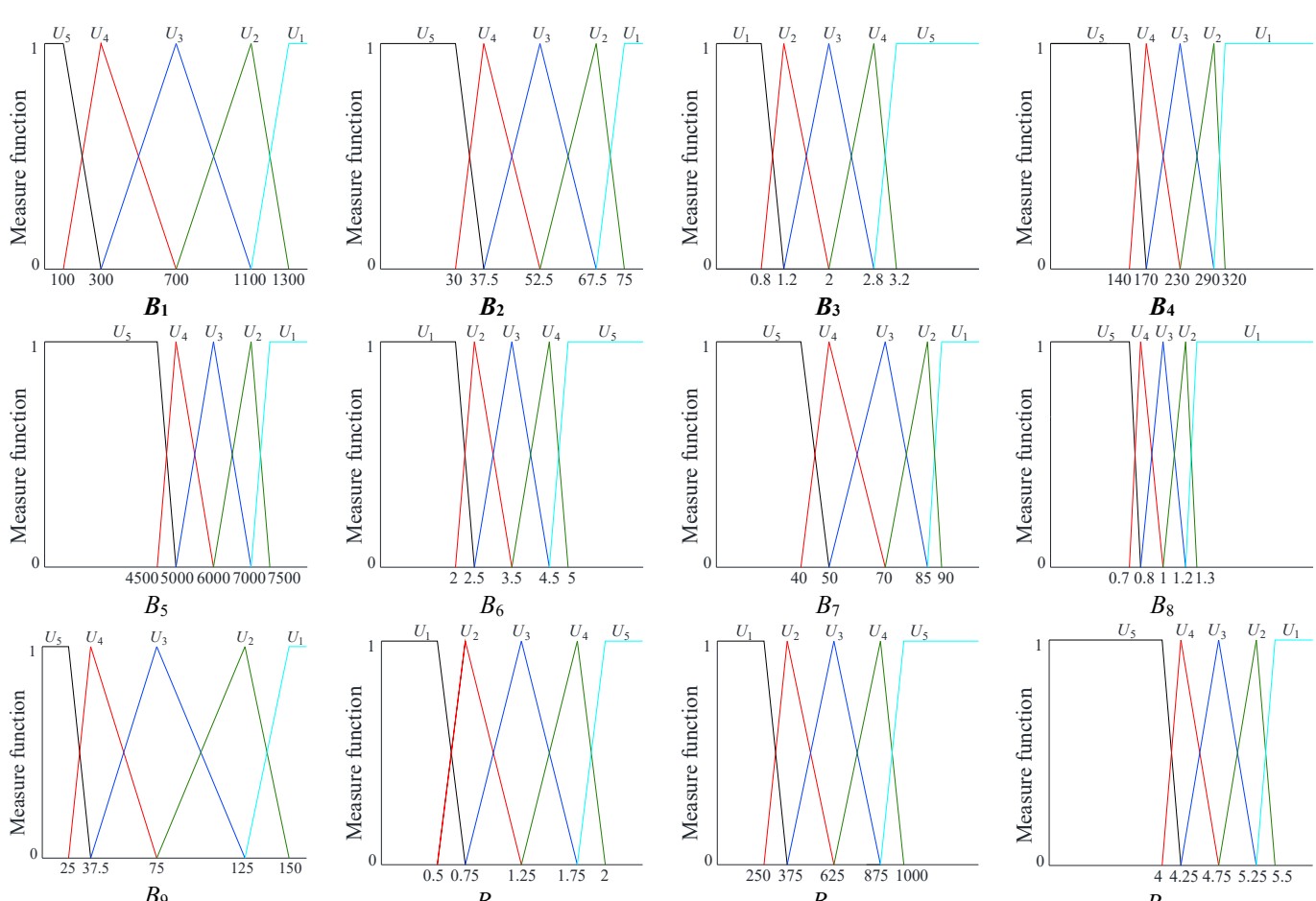

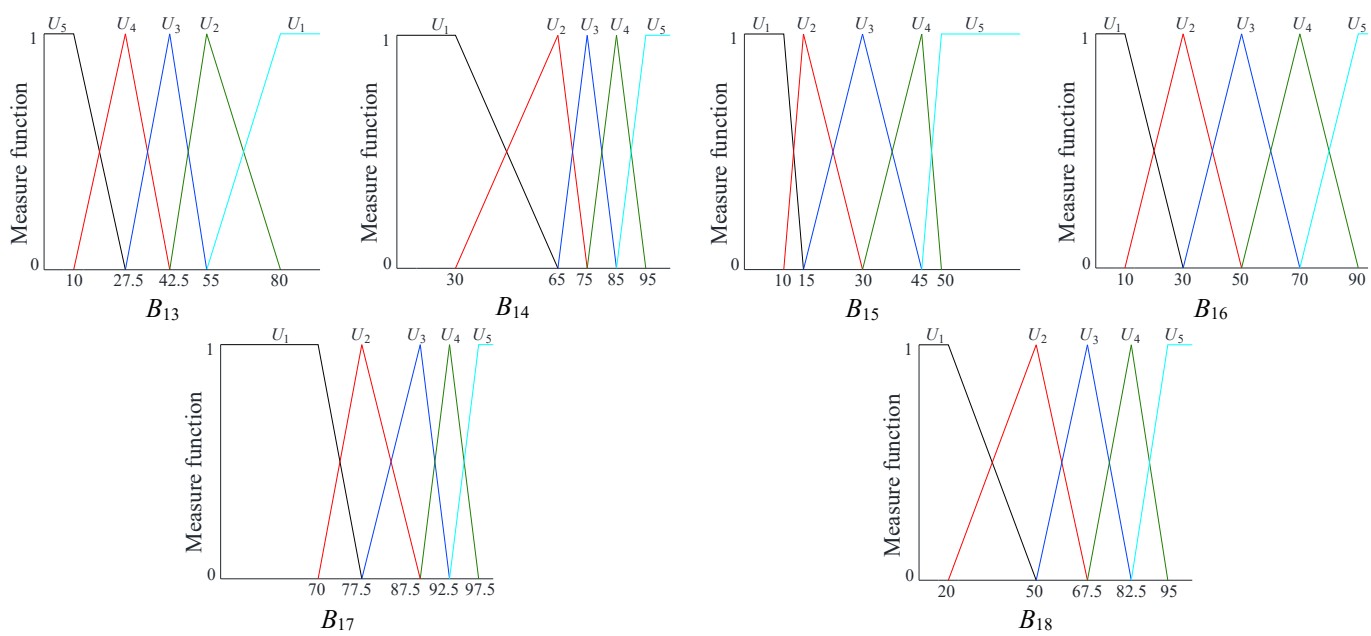

**Figure 3.** Measurement function of each indicator.

**Table 4.** WECC evaluation level and performance results.

| Year | Multiple Metrics Not Known with Certainty | | | | | Rank | Performance Index |
|---|---|---|---|---|---|---|---|
| | I | II | III | IV | V | | |
| 2015 | 0.0639 | 0.3327 | 0.4308 | 0.0833 | 0.0899 | III | 2.8044 |
| 2016 | 0.0616 | 0.2486 | 0.4038 | 0.1430 | 0.1436 | III | 3.0600 |
| 2017 | 0.0587 | 0.1359 | 0.4304 | 0.2745 | 0.1012 | III | 3.2253 |
| 2018 | 0.0587 | 0.0939 | 0.3397 | 0.3213 | 0.1870 | IV | 3.4859 |
| 2019 | 0.0587 | 0.1577 | 0.2884 | 0.3229 | 0.1729 | III | 3.3955 |
| 2020 | 0.0862 | 0.1169 | 0.2443 | 0.2548 | 0.2983 | IV | 3.5638 |

As shown in Table 4, the overall trend in the WECC of the Gansu section of the Yellow River basin gradually increased in recent years, from grade III (critical load) in 2015 to IV grade (weakly loadable) in 2020. In 2019, as its waste water discharge reached the most in recent years, and the water development and utilization rate increased abruptly compared to 2018 and 2020; it slightly decreased to Class III (critical load), but the overall trend gradually improved. In September 2019, with the introduction of the national strategy for ecological protection and high-quality development of the Yellow River basin, local governments in the Gansu Province actively respond to the national call to carry out special planning and propose corresponding measures. Until 2020, its performance index had grown from 2.804 in 2015 to 3.564 in 2020, an increase of 0.76 times, with an average annual growth rate of 4.914%. Its performance index in recent years grew year by year, and the comprehensive level had improved, but its water environment carrying capacity level is weakly loadable and still needs to be treated. Therefore, it would be necessary to continue treatment, improve the relationship between the social economy and the natural ecology, optimize the ecological environment, and improve the WECC level.

### 3.2.2. Subsystem Performance Analysis

Considering the comprehensive evaluation of the performance index, the subsystem performance was explored in depth, and a graph of the changing trends of the performance index of the indicator subsystem is established, as shown in Figure 4. By computing the performance index of each indicator subsystem in the Gansu Province from 2015 to 2020, we could visualize the changing trends of the social, economic, and natural

ecological aspects, which had gradually improved. Among them, the natural ecosystem subsystem had the highest overall performance improvement, which was 2.008 until 2020, an increase of 0.428 times over 2015, with an average annual growth rate of 4.911%. This was followed by a higher social subsystem performance index, which increased 0.22 times from 0.814 in 2015 to 1.034 in 2020, with an average annual growth rate of 4.9%. Finally, the economic subsystem reached 0.524 by 2020, an increase of 0.112 times over the initial years, with an average annual growth rate of 4.927% and the fastest growth rate among the three subsystems. This showed that in recent years, the WECC of the Gansu Province had improved, but its overall growth rate was relatively slow, which could have been due to the perennial slow economic development and poor natural environment in Gansu Province. The comprehensive performance of the economy was the lowest, and any future improvements to the economic development of the Gansu Province could increase the role of the economy in the water environment.

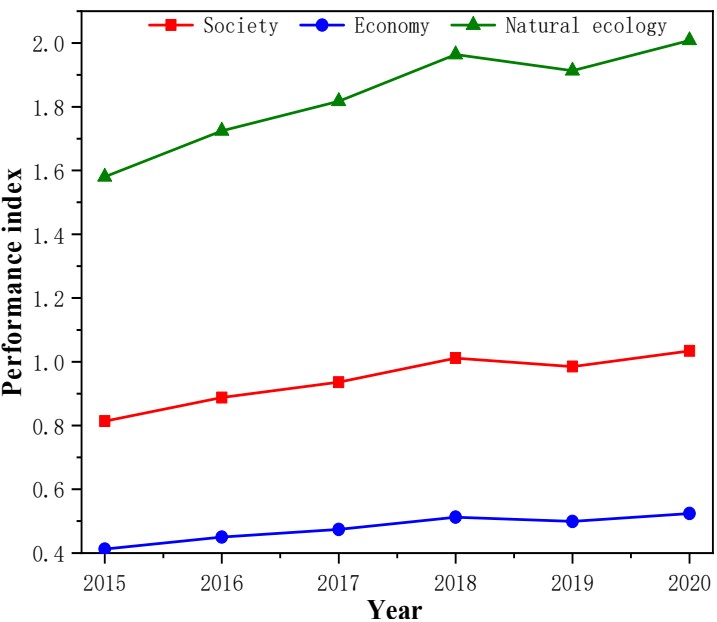

**Figure 4.** Performance index of WECC layer.

*3.3. Diagnostic Results of Disorder Factors*

The barrier degree of each index was calculated and analyzed by the barrier factor diagnosis formula (24), and the degree of influence was judged by ranking according to the degree of the barrier.

3.3.1. Diagnostic Analysis of the Guideline Layer Disorder Factor

Through the analysis of each indicator guideline layer, we explored the degree of its impact on the WECC in social, economic, and natural ecological aspects; the results are shown in Table 5. The three aspects had different trends, where the most influential was the natural ecosystem, which showed a fluctuating trend, but the overall trend was a declining state, from 65.51% in 2015 to 48.36% in 2020. The social subsystem was the next most influential, with the degree of influence rising from 2015 to 2020; the change in barrier degree rose from 17.01% in 2015 to 40.47% in 2020, while the economic subsystem had a smaller change, remaining stable, with the range of change fluctuating up and down between 10.77% and 18.14%. According to the changes in the barrier degree of these three aspects, in the period of 2015–2020, the barrier degree of waste water discharge in the Gansu Province had been at the top in recent years, and the weight of the indicator was larger, resulting in the barrier degree of the natural ecosystem being the largest, and its average value reached 55.89%. However, with the Chinese government's two rounds of

environmental protection inspectors, the Gansu Province, on the environmental rectification situation, "look back"; repeat verification; and actively carry out special studies and the development of prevention and implementation programs, so the overall decline in the barrier degree trend was more obvious. Since the beginning of 2018, the Gansu Province proposed 2018–2025 medium and long-term youth development plan, the Gansu Province Yellow River basin population density and per capita arable land area has seen a sudden increase, resulting in social subsystem barriers among the forefront, and it has become the main factor limiting the improvement of the water environment carrying capacity of the Gansu section of the Yellow River basin. Until 2020, its barrier degree reaches 40.47%, and the overall trend is plummeting, which is expected to become the biggest constraint in the coming years, and it needs to pay more attention to it and take certain measures so as to reduce its barrier degree.

**Table 5.** Subsystem barrier degree (%).

| Year | Subsystem Barrier Degree (%) | | |
|---|---|---|---|
| | **Social Subsystems** | **Economic Subsystem** | **Natural Ecosystem Subsystem** |
| 2015 | 17.01 | 17.48 | 65.51 |
| 2016 | 20.80 | 11.36 | 67.85 |
| 2017 | 33.05 | 18.14 | 48.81 |
| 2018 | 33.74 | 16.24 | 50.02 |
| 2019 | 34.45 | 10.77 | 54.78 |
| 2020 | 40.47 | 11.17 | 48.36 |

3.3.2. Diagnostic Analysis of Indicator Layer Disorder Factors

The results of detailed analysis of the barrier degree of each index in the Yellow River basin Gansu section from 2015 to 2020 are shown in Figure 5. The obstacle factors were distributed in the natural ecosystem each year, and the obstacle factors in 2015–2017 were water quality excellence, water resource development and utilization rate, wastewater discharge, and per capita arable land area. In 2018, the proportion of major barrier factors in the social subsystem was relatively high, and population density and urbanization rate were among the top major barriers that limited the improvement of the WECC of the Gansu section of the Yellow River basin. In 2019–2020, the main obstacle factors were distributed in the natural ecological subsystem and the social subsystem, including wastewater discharge, population density, urbanization rate, and forest cover, with the factors in the social subsystem greatly increasing the constraints on the WECC, as compared to the previous years. This indicated that the natural ecological aspects had been managed to a certain extent, but the social factors lacked the corresponding oversight.

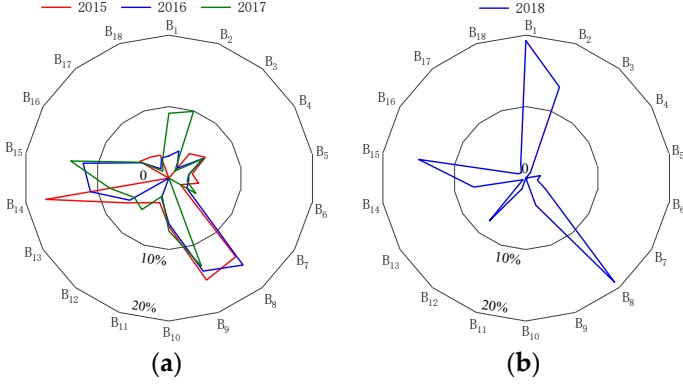

(a)　　　　　　　　　　(b)

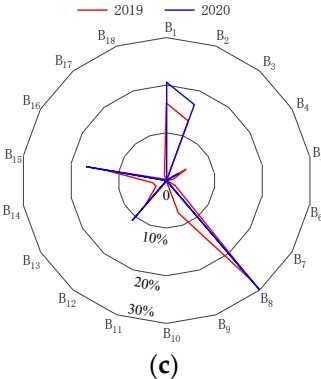

(**c**)

**Figure 5.** WECC indicator layer barrier degree: (**a**) 2015-2017 barrier degree of indicator layer; (**b**) 2018 barrier degree of indicator layer; (**c**) 2019-2020barrier degree of indicator layer.

## 4. Conclusions

Based on the conceptual framework of SENCE, a WECC index system was constructed for the Gansu section of the Yellow River Basin and 18 index systems were determined according to social–economic–natural ecological aspects. The subjective and objective assignments were identified by an improved G1-CRITIC method, and the weights were assigned by the moment estimation theory, and then they were diagnosed and evaluated according to the unconfirmed measurement theory and obstacle factor model. Lastly, the WECC was derived according to the evaluation grade and obstacle degree results. The research findings were as follows.

(1) Using the unconfirmed measurement model, we found that the WECC value of the Gansu section of the Yellow River basin from 2015 to 2020 was slowly increasing, from grade III (critical load) to IV (weakly loadable), but slightly decreasing to grade III (critical load) in 2019, with the overall trend gradually improving. Its comprehensive performance level also fluctuated and increased, from 2.804 in 2015 to 3.564 in 2020, an increase of 0.76, with an average annual growth rate of 4.914%, of which the natural ecological subsystem had the highest performance index, growing to 2.008 by 2020, an increase of 0.428 times. The social subsystem had the fastest growth with an average annual growth rate of 4.927%. All the subsystems had been slowly improving, but they still have room for improvement in the future.

(2) According to the diagnostic model of obstacle factors to diagnose the obstacle degree of each indicator, the natural ecological subsystem was the most influential factor in the criterion layer, but the overall trend was fluctuating and decreasing, from 65.51% in 2015 to 48.36% in 2020, while the social subsystem had the second highest degree of influence but showed a rising trend year by year. In the indicator layer, the biggest obstacle factors in the last three years varied from other years, and wastewater discharge, population density, urbanization rate, and forest coverage rate were the most significant obstacle factors, which were concentrated in the natural ecological and social subsystems.

(3) The unconfirmed measure theory was used to evaluate the WECC of the Gansu section of the Yellow River basin, and the key impact factors were diagnosed based on the barrier factor model.

This study had some limitations, as the detailed division of the cities in the Gansu section of the Yellow River basin was not studied. This study analyzed three aspects, social, economic, and natural ecological, but the water quality of water bodies was not examined in detail, and further exploration of these aspects will be needed in future research.

**Author Contributions:** Conceptualization, C.J. and Q.G.; Investigation, Y.Z. and Z.J.; Methodology and Formal Analysis, C.J. and Q.G.; Writing—Original Draft Preparation, L.G. and Q.G.; Writing—

Review and Editing, C.J. and Q.G.. All authors have read and agreed to the published version of the manuscript.

**Funding:** This study was the financially supported by the National Science Foundation (51969011) of China, Gansu Provincial Science and Technology Program (20JR10RA274).

**Acknowledgments:** The authors gratefully acknowledge many important contributions from the researchers of all reports cited in our paper.

**Conflicts of Interest:** The authors declare no conflict of interest.

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
