# Peer review of "Evaluation of Regional Water Environmental Carrying Capacity and Diagnosis of Obstacle Factors Based on UMT Model"

_water, doi:10.3390/w14172621_

Round 1

Reviewer 1 Report

The work is structured correctly.

The applied methodology is correctly explained and the schemes are clear.

Although this is a mainly statistical work applied to an example territory, it would be appropriate to contextualise the area in question in a more precise way.

In this way the reader is able to better understand the example in the studio.

It is therefore advisable to include a figure that represents the area in question and basic information (area, main inhabited centers, population, etc.

Tab. 1 layout so as not to interrupt the paragraphs between one page and another.

Author Response

Please see our response in the attachment file, thank you very much for your professional review.

Reviewer 2 Report

Evaluation of Regional Water Environmental Carrying Capacity and Diagnosis of Obstacle Factors based on UMT Model

In the paper the Water Environmental Carrying Capacity WECC is studied, taking the Gansu section of the Yellow River basin as the research object using data from 2015 to 2020. Based on the conceptual framework of SENCE, a WECC index system was constructed according to social, economic and natural ecological aspects. The subjective and objective assignments were identified by an improved G1-CRITIC method, and the weights were assigned by the moment estimation theory, and then they were diagnosed and evaluated according to the unconfirmed measurement theory and obstacle factor model. Lastly, the WECC was derived according to the evaluation grade and obstacle degree results.

In general terms, the article presents an interesting work and its methodology contains innovative elements. although the structure in which the information has been presented makes it difficult to read. The analysis of results should be more solid, supporting the statements made with data and relying on scientific literature. I would recommend that the article be restructured so that it has the typical structure of the scientific article in a format as requested by Water.

Some specific aspects are detailed below.

The meaning of the UMT model is not explicit. It is included in the title but not detailed in the body of the article

Line 37: The paragraph “In China, General Secretary Xi Jinping has visited the provinces and regions along the Yellow River several times and elevated the ecological protection” should be reformulated. It is not necessary to quote Xi Jinping, it should refer to the Government of the People's Republic of China.Line 46 DPSIRM Se debe detallar los acrónimo la primera vez que se citan 

Line 67 SENCE, the equivalence of the acronyms must be described the first time they are cited

Figure 3 is included but in the methodology, the criteria for the elaboration of figure 3 are not detailed and an analysis of said figure is not presented, please include these aspects

Line 281-281 “which could have been due to the perennial slow economic development and poor natural environment in Gansu Province”. In item 3.1 Study area, you should give more details, figures, that allow you to justify this statement.

Line 248 Item 3.3.2 WECC evaluation level and comprehensive performance  

The observed trend and fluctuations of the WECC of the Yellow River Basin between 2015 and 2020 are shown, but it is not detailed whether these fluctuations are due to implemented policies, economic effects, natural or climatic factors, etc. If, in general terms, an improvement is evident (from grade III, critical load to IV,weakly loadable), it is important to know clearly what produced said improvement.

Something similar happens with the Diagnostic analysis of the guideline layer disorder factor (Line 289), it is requested to improve the discussion of results.

Round 2

Reviewer 2 Report

Suggestions have been accepted. I think the article can be published